# A Comprehensive Analysis of Chloroplast Genome Provides New Insights into the Evolution of the Genus *Chrysosplenium*

**DOI:** 10.3390/ijms241914735

**Published:** 2023-09-29

**Authors:** Tiange Yang, Zhihua Wu, Jun Tie, Rui Qin, Jiangqing Wang, Hong Liu

**Affiliations:** 1Hubei Provincial Key Laboratory for Protection and Application of Special Plant Germplasm in Wuling Area of China, College of Life Sciences, South-Central Minzu University, Wuhan 430074, China; yangtge@163.com (T.Y.); tiejun@mail.scuec.edu.cn (J.T.); qinrui@scuec.edu.cn (R.Q.); 2College of Life Sciences, Zhejiang Normal University, Jinhua 321004, China; zhuawu@zjnu.edu.cn; 3College of Computer Science, South-Central Minzu University, Wuhan 430074, China

**Keywords:** *Chrysosplenium*, chloroplast, selection pressure, phylogeny, comparative genome

## Abstract

*Chrysosplenium*, a perennial herb in the family Saxifragaceae, prefers to grow in low light and moist environments and is divided into two sections of *Alternifolia* and *Oppositifolia* based on phyllotaxy. Although there has been some progress in the phylogeny of *Chrysosplenium* over the years, the phylogenetic position of some species is still controversial. In this study, we assembled chloroplast genomes (cp genomes) of 34 *Chrysosplenium* species and performed comparative genomic and phylogenetic analyses in combination with other cp genomes of previously known *Chrysosplenium* species, for a total of 44 *Chrysosplenium* species. The comparative analyses revealed that cp genomes of *Chrysosplenium* species were more conserved in terms of genome structure, gene content and arrangement, SSRs, and codon preference, but differ in genome size and SC/IR boundaries. Phylogenetic analysis showed that cp genomes effectively improved the phylogenetic support and resolution of *Chrysosplenium* species and strongly supported *Chrysosplenium* species as a monophyletic taxon and divided into three branches. The results also showed that the sections of *Alternifolia* and *Oppositifolia* were not monophyletic with each other, and that *C. microspermum* was not clustered with other *Chrysosplenium* species with alternate leaves, but with *C. sedakowii* into separate branches. In addition, we identified 10 mutational hotspot regions that could serve as potential DNA barcodes for *Chrysosplenium* species identification. In contrast to *Peltoboykinia*, the *clp*P and *ycf*2 genes of *Chrysosplenium* were subjected to positive selection and had multiple significant positive selection sites. We further detected a significant positive selection site on the *pet*G gene between the two sections of *Chrysosplenium*. These evolutionary characteristics may be related to the growth environment of *Chrysosplenium* species. This study enriches the cp genomes of *Chrysosplenium* species and provides a reference for future studies on its evolution and origin.

## 1. Introduction

The chloroplast (cp) genome has long been a major source of molecular data for studying plant phylogeny and evolution because of its maternally inherited and relatively conserved nature. The size and structure of cp genomes have been highly conserved during land plant evolution, in contrast to the large variation in the size and structure of plant mitochondrial genomes. In Saxifragaceae species, the cp genome has a highly conserved circular quadripartite structure containing a large single-copy (LSC) and a small single-copy (SSC) divided by two inverted repeat (IR) regions. With the development of sequencing technology, whole-genome sequencing and large-scale phylogenetic analysis of cp genomes of most plants have been achieved, further facilitating plant taxonomic studies [1,2].

*Chrysosplenium* L., belonging to the family Saxifragaceae, is a small perennial herbaceous plant, usually with flagellate branches or bulbs, whose phyllotaxy is divided into alternate and opposite leaves. There are about 80 *Chrysosplenium* species in the world, which are mainly distributed in Asia, Europe and North America in the Northern Hemisphere, and a few in temperate regions in the Southern Hemisphere, mainly two species located in and around Chile, namely *C. valdivicum* and *C. macranthum* [3,4]. In China, there are about 38 species and 15 varieties, accounting for more than 56% of the total number of *Chrysosplenium* species in the world, of which 23 species are endemic to China, mainly in northern and southern China [5,6,7]. In addition, the shade-loving and moisture-loving characteristics of *Chrysosplenium* species make them ideal materials for studying the evolution of low-light and low-temperature adaptations in plants. Taxonomic studies on *Chrysosplenium* can be traced back as far as the mid-18th century when *C. alternifofium* L. with alternate leaves and *C. oppositifolium* L. with opposite leaves were recognized by Linnaeus (1753). Subsequently, at the end of the 19th century, some species were added to *Chrysosplenium* and classified accordingly [8,9]. In 1877, Maximowicz et al. (1877) divided the *Chrysosplenium* into subgen. *Gamosplenium* and subgen. *Dialysplenium* based on the length of the sepals and stamens [8]. In 1890, the *Chrysosplenium* was divided into the groups of *Alternifolia* and *Oppositifolia* based on opposite and alternate leaves [9]. In 1957, Hara (1957) made a detailed morphological study of *Chrysosplenium* and identified 55 species divided into sections of *Alternifolia* and *Oppositifolia* [3]. In 1986, Pan (1986) identified and studied the *Chrysosplenium* species in China and classified them into two subgenera (*Chrysosplenium* and *Gamosplenium*), as well as five groups and ten lineages [10,11]. Since then, many new *Chrysosplenium* species have been discovered, and the diversity of *Chrysosplenium* species has been continuously enriched [6,7,12,13]. 

Previous phylogenetic studies of *Chrysosplenium* have mainly used cp fragments and nuclear ribosomal DNA (nrDNA) sequences, while the cp genome has been relatively little studied, and the phylogenetic position of a few species was still controversial. Nakazawa et al. (1997) evaluated the phylogeny of *Chrysosplenium* species using *rbc*L and *mat*K sequences and found that *mat*K sequences had a high phylogenetic resolution [14]. Soltis et al. (2001) studied the phylogeny of some *Chrysosplenium* species based on *mat*K genes and showed that the sections of *Alternifolia* and *Oppositifolia* were monophyletic sisters (Appendix A) [15]. This phylogeny has long been in common use. Afterwards, Xiang et al. (2012) performed a phylogenetic analysis of *Saniculiphyllum* based on four chloroplast DNA (*trn*L-*trn*F, *psb*A-*trn*H, *mat*K, *rbc*L) and two nrDNA fragments (nrITS, rrn26S) [16]. In the *Chrysosplenium* branch, *C. microspermum* with alternate leaves clustered with *C. nepalense* with opposite leaves (Appendix A). Tkach et al. (2015) investigated the phylogeny of *Micranthes* based on nrITS and *trn*L-*trn*F sequences [17]. In the *Chrysosplenium* branch, *C. microspermum* was located at the base of the *Chrysosplenium* branch (Appendix A). In the same year, Deng et al. (2015) studied the phylogeny and evolutionary history of *Chrysosplenium* based on the matrices of cpDNA and nrDNA [18]. The cpDNA-based BI tree showed that *Chrysosplenium* was mainly divided into three clades, and *C. microspermum* was located at the base of the *Chrysosplenium* branch (Appendix A). The ML tree based on the matrices of cpDNA and nrDNA showed similar results, but the nucleoplasmic-based BI tree showed that *Chrysosplenium* was divided into two branches corresponding to the sections of *Alternifolia* and *Oppositifolia*, and that *C. microspermum* was clustered in the section *Alternifolia* branch (Appendix A). Subsequently, Folk et al. (2019) performed a phylogenetic analysis of 627 Saxifragales species based on 301 protein-coding loci, in which *Chrysosplenium* species were divided into three branches, with the sections of *Alternifolia* and *Oppositifolia* not being monophyletic sisters of each other, *C. microspermum* with alternate leaves clustered in the section *Oppositifolia* branch, and *C. sedakowii* with alternate leaves forming a separate branch (Appendix A) [19]. To date, the phylogenetic position of *C. microspermum* has not been clarified.

With the publication of the *C. aureobracteatum* cp genome in 2018 [20], studies on the cp genome of *Chrysosplenium* were gradually initiated. Then, the six cp genomes of *Chrysosplenium* species revealed cp genome characteristics of *Chrysosplenium* [4]. Subsequently more cp genomes of *Chrysosplenium* species were published [21,22]. Nevertheless, the cp genomes of many *Chrysosplenium* species are still unknown. Therefore, in order to gain a comprehensive understanding of the phylogenetic relationships among *Chrysosplenium* species, this study first de novo assembled and annotated the cp genomes of 34 *Chrysosplenium* species. Together with the published species, a total of 44 *Chrysosplenium* cp genomes were further performed for comparative genomics and phylogenetic analysis. The primary research questions addressed in this study are as follows. (1) Whether the sections of *Alternifolia* and *Oppositifolia* are monophyletic sisters to each other in the phylogeny of the 44 *Chrysosplenium* species? (2) Where is the phylogenetic location of *C. microspermum*? (3) Are there significant differences in the cp genomes of *Chrysosplenium* between species and between the two groups? (4) Is *Chrysosplenium* under significant positive selection on the cp genome compared to *Peltoboykinia*? 

## 2. Results

### 2.1. Structural Characterization of the Chloroplast Genome of Chrysosplenium

All cp genomes of the 44 *Chrysosplenium* species presented a typical quadripartite structure with a large single-copy (LSC), a small single-copy (SSC), and two inverted repeats (Ira and Irb). The size of cp genome ranged from 148,566 bp to 154,441 bp, with an average of 152,576 bp (Figure 1; Appendix A). The GC content of cp genomes ranged from 37.22% to 37.72%, with an average of 37.45%. Among the protein-coding genes (PCGs), 5 genes were responsible for photosystem I (*psa*A, *psa*B, *psa*C, *psa*I, *psa*J), 15 genes for photosystem II (*psb*A, *psb*B, *psb*C, *psb*D, *psb*E, *psb*F, *psb*H, *psb*I, *psb*J, *psb*K, *psb*M, *psb*N, *psb*T, *psb*Z, *ycf*3), 6 genes for ATP synthase (*atp*A, *atp*B, *atp*E, *atp*F, *atp*H, *atp*I), 9 genes for large ribosomal proteins (*rpl*2, *rpl*14, *rpl*16, *rpl*20, *rpl*22, *rpl*23, *rpl*32, *rpl*33, *rpl*36), and 12 genes for small ribosomal proteins (*rps*2, *rps*3, *rps*4, *rps*7, *rps*8, *rps*11, *rps*12, *rps*14, *rps*15, *rps*16, *rps*18, *rps*19) were found in *Chrysosplenium* (Appendix A). In addition, we found that some PCGs were lost to varying degrees, such as *rpl*32, *ndh*A, *ndh*F, and *ndh*G. Interestingly, *rpl*32 was only annotated in some *Oppositifolia* species, *ndh*A was missing in both *C. carnosum* and *C. forrestii*, and *ndh*G and *ndh*F were only missing in *C. carnosum* (Appendix A).

### 2.2. Repeat Identification

The MISA v. 1.0 software was utilized to detect simple sequence repeats (SSR) in 44 cp genomes of *Chrysosplenium* (Figure 2a; Appendix A). The results of SSR analysis revealed a variation in the number of SSRs, ranging from 75 to 150. These SSRs were predominantly located in the LSC and SSC regions of the gene spacer Among the six types of SSRs, the largest number was dinucleotide repeats, accounting for 36.7%, followed by mononucleotide and tetranucleotide repeats, accounting for 25.5% and 22.9%, respectively. The smallest number was hexanucleotide repeats, accounting for only 0.85%. We examined the number and distribution of long repeats in the cp genomes of 44 *Chrysosplenium* species, which ranged from 19 to 50, with an average of 31 repeats, mainly in the IR and LSC regions (Figure 2b; Appendix A). Fourteen *Chrysosplenium* species contained only forward and palindromic repeats, namely *C. uniflorum*, *C. henryi*, *C. glossophyllum*, *C. zhouzhiense*, *C. flagelliferum*, *C. nudicaule*, *C. hydrocotylifolium*, *C. echinus*, *C. nepalense*, *C. kiotense*, *C. lanuginosum*, *C. delavayi*, *C. aureobracteatum*, and *C. macrospermum*.

### 2.3. Divergence Hotspots and Rearrangement Analysis

To evaluate the differences in cp genomes among 44 *Chrysosplenium* species, we performed mVISTA analysis with the annotated *C. ramosum* cp genome as a reference (Appendix A). The cp genomes of the 44 *Chrysosplenium* species showed relatively similar patterns, with the main sequence variations observed in the non-coding regions. On the other hand, the exons and untranslated regions (UTR) exhibited minimal variation across genomes. Nucleotide diversity analysis revealed that coding regions were more conserved than non-coding regions. Among these hot spots, eight intergenic regions (IGSs) (*trn*S-GCU-*trn*G-UCC, *atp*H-*atp*I, *rpo*B-*trn*C-GCA, *psa*A-*ycf*3, *ndh*C-*trn*V-UAC, *acc*D-*psa*I, *ycf*4-*cem*A, *ndh*F-*rpl*32) and two genes (*mat*K, *ycf*1) showed the highest levels of divergence (Figure 3). Rearrangement analysis indicated that the cp genomes of 44 *Chrysosplenium* species were relatively conserved, and no significant rearrangements were found (Appendix A). Intraspecific variation also exists in the genus *Chrysosplenium*, mainly in the spacer region, e.g., C. sinicum (Appendix A).

### 2.4. Dynamic Analysis of the IR Boundary

We analyzed the dynamics of the IR boundaries of the cp genomes of 44 *Chrysosplenium* species. The boundary situation is different for some species, and the expansion and contraction of the IR regions leads to changes in the cp genes at the IR boundaries, with some genes entering the LSC and SSC regions. The results showed that the four boundaries of the cp genomes of 44 *Chrysosplenium* species were relatively conserved (Figure 4). The *rps*19 genes of *C. pilosum*, *C. microspermum*, and *C. aureobracteatum* were located in the LSC region, and the *rps*19 genes of other 41 species were located in the LSC-IRb boundary. The vicinity of the IRb-SSC boundary mainly contained *trn*N and *ndh*F genes. The *trn*N genes were present in IRb in all *Chrysosplenium* species, while the *ndh*F genes of *C. ramosum*, *C. biondianum*, *C. uniflorum*, and *C. forrestii* were exclusively located in the SSC region. The *ndh*F genes of the other 40 *Chrysosplenium* species were located on the IRb-SSC boundary. The SSC-IRa boundary had *ycf*1 and *trn*N genes, with the *ycf*1 gene located on the boundary and the *trn*N gene located in the IRa region. Near the SSC-IRa boundary, there were also *rpl*2 and *trn*H genes, with the *rpl*2 gene located in the IRa region and the *trn*H gene located in the LSC region. This phenomenon is similar in all *Chrysosplenium* species.

### 2.5. Codon Usage Analysis

We selected 53 PCGs (>300 bp) for codon usage analysis. Codon analysis revealed some differences in codon usage numbers, GCs, and GC3s among the 44 *Chrysosplenium* species (Figure 5a; Appendix A). The *C. forrestii* and *C. carnosum* had lower codon numbers, with clade B showing greater variation in codon numbers than the clade C branch, which was overall more stable. Leucine was found to be the most abundant amino acid in the cp genome, while cysteine was relatively rare. Among the 61 codons, AAU encoded the most frequent occurrence of isoleucine and UGC encoded the least frequent occurrence of cysteine. The trend in GCs and GC3s was generally consistent and lower in 44 *Chrysosplenium* species than in *P. tellimoides*. Clade B generally had higher levels of GCs and GC3s than the C branch. Relative synonymous codon usage (RSCU) values of the 44 species were similar, with 61 codons encoding 20 amino acids (Figure 5b; Appendix A). The RSCU value for serine encoded by AGC was the lowest, while leucine encoded by UUA had the highest RSCU value. Both tryptophan (UGG) and methionine (AUG) were encoded by only one codon and had RSCU values of one. Furthermore, twenty-nine codons had RSCU values greater than one, indicating biased use.

### 2.6. Selective Pressure Analyses

We analyzed selection pressure on the 44 *Chrysosplenium* species and *P. tellimoides*, with a total of 990 combinations. This result showed that the *Chrysosplenium* species were not subject to positive selection in the species level (Appendix A; Appendix A). In the gene level, the LSC region had more positively selected genes (PSGs) than the SSC and IR regions in the sections of *Alternifolia* and *Oppositifolia* (Figure 6 and Appendix A; Appendix A). PSGs in the LSC region included *atp*F, *mat*K, *ndh*J, *psa*I, *psb*K, *psb*L, *rpl*33, *rps*11, *rps*14, *rps*16, *rps*18, *rps*2, and *rps*8 (Figure 6a); PSGs in the IR region included *ndh*B and *rps*12 (Figure 6b). PSGs in the SSC region included *ccs*A, *ndh*E, and *rps*15 (Figure 6c); The remaining genes were generally subject to purifying selection. In addition, other PSGs were detected in the *Alternifolia* and *Oppositifolia*, respectively (Appendix A), such as: *acc*D, *atp*E, *atp*I, *cem*A, *clp*P, *ndh*C, *ndh*K, *pet*A, *pet*D, *psa*J, *rbc*L, *rpl*20, *rpl*22, *rpo*A, *rps*3, *rps*4, *ycf*3, *ycf*4, *ccs*A, *ndh*D, *ndh*E, *ndh*H, *ndh*I, *rps*15, *ycf*1, *ndh*B, *rps*12, and *ycf*2. Compared to *P. tellimoides*, the *ycf*2 and *clp*P genes were positively selected in 44 *Chrysosplenium* species and had multiple significant positive selection sites, and we also detected one significant positive selection site in the *pet*L gene (Figure 6d, Figure 7b–d and Appendix A; Appendix A). These PSGs may be associated with the adaptation of *Chrysosplenium* species to low-light and low-temperature environments. Furthermore, we did not detect a PSG that could significantly distinguish the two sections in the genus *Chrysosplenium*, but we detected a significant positive selection site on the *pet*G gene, which may be related to the differential evolution of the two sections (Figure 7a; Appendix A).

### 2.7. Phylogenetic Analysis

The cpPCGs matrix length was 72,828 bp, including 8401 parsimony informative sites, 16,649 variable sites and 52,882 conserved sites. The nrDNA matrix length was 6854 bp, including 926 parsimony informative sites, 1326 variable sites and 5169 conserved sites. The cpPCGs + nrDNA matrix length was 79,682 bp, including 9325 parsimony informative sites, 17,974 variable sites and 58,052 conserved sites (Appendix A). Phylogenetic trees of the three matrices were constructed by the Maximum Likelihood and Bayesian Inference methods, respectively. The phylogenetic trees of both cpPCGs matrix and cpPCGs + nrDNA matrix have high confidence, while the nrDNA matrix phylogenetic tree as a whole has some branches with low support, which was significantly different from the phylogenetic trees of cpPCGs matrix and cpPCGs + nrDNA matrix (Appendix A). The topology of the phylogenetic tree of cpPCGs + nrDNA matrix obtained by the two methods was similar, and most of the nodes had high support rates and posterior probabilities (Figure 8, Appendix A). The phylogenetic tree of cpPCGs + nrDNA matrix showed that *Chrysosplenium* species were more closely related to *Peltoboykinia* in the family Saxifragaceae. Three branches were formed within the *Chrysosplenium*, with *C. microsperm* and *C. sedakowii* with alternate leaves forming clade A alone, other alternate leaf species forming clade B, and opposite leaf species forming clade C. Clade A and clade B correspond to the *Alternifolia*, while clade C corresponds to the *Oppositifolia*, indicating that the two sections were not monophyletic. Clade B can be divided into three subclades, with species in subclade B1 generally distributed at low and middle altitudes, species in subclade B2 more widely distributed, and species in subclade B3 generally distributed at high altitudes in China. Clade C was mainly divided into two subbranches, with species in subclade C1 more widely distributed (e.g., Europe, North America, South America, and Asia), and subclade C2 species originate mainly from the northeastern regions of East Asia (northeastern China, Korea, North Korea, and Japan).

## 3. Discussion

### 3.1. Chloroplast Genome Evolution within Chrysosplenium Species

Our study analyzed the cp genomes of 44 *Chrysosplenium* species and found that they were not highly variable in size. The genomes were conserved in terms of structure, gene composition, and gene order, similar to many angiosperm genera. The distribution of GC content in the cp genome was uneven, with the highest GC content in the IR region and the lowest in the SSC region. The presence of four rRNAs (rrn4.5, rrn5, rrn16, rrn23) in the IR region may lead to the higher GC content [23,24,25]. Additionally, the expansion and contraction of the IR region also played a role in changing the size of the cp genome and the boundary genes. Comparison of the IR/SC boundaries among 44 *Chrysosplenium* species revealed high similarity. While the boundary region of the cp genome was relatively stable, the expansion and contraction of the IR region may lead to alterations in the *ndh*F and *ycf*1 genes located in the boundary region. It was observed that cp genes are rarely lost and are likely transferred to the nuclear genome or functionally replaced by nuclear genes [26,27,28]. Interestingly, we found that *ndh*A was lost in *C. forrestii* and *C. carnosum*, while *ndh*F and *ndh*G were lost in *C. carnosum*. Apart from the deletion of individual genes, no other significant variation was found. The deletion of genes may be related to the environment, such as the loss of the NDH gene in some orchids [29]. Differences between genomes also exist within species, a common phenomenon that may be due to genetic variation and geographic distribution during the evolution of the species. Similar differences exist in the genus *Chrysosplenium*, where sequence alignment revealed differences in the cp genomes of two different taxa of *C. sinicum*, mainly in the spacer region. However, there is still a lack of a more comprehensive resolution of genomic differences among intraspecific species in the genus *Chrysosplenium*.

A comparable number of SSRs and long repeats were identified in 44 different *Chrysosplenium* species. However, the types of SSRs and long repeats differed among the species. These repeats were predominantly found in the intergenic spacer (IGS) of the large single copy (LSC) region. Mononucleotide and dinucleotide repeats were the most common types of SSRs, while forward and palindromic repeats were the predominant types of long repeats. Since the Pi value of PCGs and IGSs was highest on average in IR, LSC and SSC, we found that *mat*K, *trn*S-GCU-*trn*G-UCC, *acc*D-*psa*I, *ycf*1, *ndh*F-*rpl*32, *atp*H-*atp*I, *rpo*B-*trn*C-GCA, *psa*A-*ycf*3, *ycf*4- *cem*A, and *ndh*C-*trn*V-UAC had high Pi values and were candidate markers to distinguish *Chrysosplenium* species, but further experimental studies were needed for specific conclusions.

### 3.2. Selection Pressure Analysis of Chrysosplenium Species

Species grow in various environments and are often influenced by different climatic factors, such as humidity, light, altitude and temperature. Some genes may be subject to positive selection in response to environmental changes. Our results indicate that the majority of genes exhibited an average *K*a/*K*s ratio below one. Purifying selection, a prominent mechanism of natural selection, plays a crucial role in continuously removing harmful mutations. These genes hold significant importance in facilitating plant adaptation and ensuring survival. Positive selection is usually associated with adaptive traits. *Chrysosplenium* species have a wide range of altitudinal distribution, both at low and high altitudes, and prefer shady and humid environments. In *Chrysosplenium*, most genes were found to be under purifying selection, and only a small number of genes were under positive selection across species, so the purifying selection of most cp genes may be the result of their adaptive evolution. No significant PSGs were detected in combinations of the *Oppositifolia* and *Alternifolia* species, and only some combinations were detected to contain PSGs. In terms of the PSG number, the LSC region was more numerous than the SSC and IR regions. Among them, *psb*L, *rps*18, *ndh*B, and *rps*12 genes showed strong positive selection in most species. Additionally, we detected the *pet*G genes in both the *Oppositifolia* and *Alternifolia* to contain a significant locus despite not being under positive selection. The *pet*G genes are primarily associated with photosynthesis [30], suggesting that there may be some differences in photosynthesis between the two sections. The *P. tellimoides*, *ycf*2 and *clp*P genes were subjected to significant positive selection in almost 44 *Chrysosplenium* species. In angiosperms, the *ycf*2 genes were susceptible to positive or purifying selection [31,32]. Although the exact function and role of *ycf*2 remains unclear, studies have shown that *ycf*2 genes were associated with photosynthesis, leaf patterning, cell survival and ATPase metabolism [33,34,35,36]. The positive selection of *ycf*2 genes indicated that this gene may be involved in the evolution of low-light adaptations in *Chrysosplenium* species. The *clp*P gene encoding *clp*P protease is also subject to positive selection in some angiosperms, such as *Paphiopedilum* (Orchidaceae) [37], *Acacia* (Fabaceae) [38], *Bupleurum* (Apiaceae) [39], and *Ficus* (Moraceae) [40], and shows high variability in Amaryllidaceae and Papilionoideae [41,42], suggesting that it may accelerate substitution rates in some angiosperms. The *clp*P protease can degrade or repair damaged proteins [43] and is important for plant development in response to environmental changes [44]. Thus, the positive selection of *clp*P gene may help *Chrysosplenium* species to adapt to low light and low temperature environments. In summary, these PSGs may contribute to the adaptation of different *Chrysosplenium* species to different environments and can be used as candidate genes to further investigate the adaptive evolutionary mechanism.

### 3.3. Phylogenetic Relationships of Chrysosplenium Species

The Maximum Likelihood and Bayesian Inference methods were used to construct phylogenetic trees for 44 *Chrysosplenium* species, and the topology of the phylogenetic trees obtained by these two methods was similar. The phylogenetic trees of cpPCGs and cpPCGs + nrDNA matrices had a similar structure with strong support, the use of nrDNA sequences alone was not well supported for some species, and there was a clear inconsistency between nucleoplasm. Nevertheless, these results suggest that *Chrysosplenium* was monophyletic, which was supported by previous studies [4,14,15,19,20]. Furthermore, our results showed the division of *Chrysosplenium* into three main clades, corresponding to the sections of *Alternifolia* (clade A and clade B) and *Oppositifolia* (clade C). Clade A (*C. microspermum* and *C. sedakowii*) with alternate leaves was located at the base of the *Chrysosplenium* branch, suggesting that it had a comparable evolutionary position in *Chrysosplenium*, but this was somewhat at variance with previous studies. Soltis et al. (2001) studied the phylogenetic relationships of some *Chrysosplenium* species based on *mat*K sequences and showed that *Chrysosplenium* was divided into two mutually monophyletic branches [15]. However, some *Chrysosplenium* species such as *C. microspermum* and *C. sedakowii* were lacking in this study, and the phylogeny of *Chrysosplenium* was still not clear enough. A small number of cpDNA and nrITS markers were not sufficiently stable for the phylogenetic position of *C. microspermum*. Phylogeny using four chloroplast DNA and nrDNA markers showed *C. microspermum* clustered into a clade with opposite leaf species (*C. nepalense*), while phylogeny based on nrITS and *trnL-trnF* markers indicated that *C. microspermum* was located at the base of the *Chrysosplenium*. In Deng et al. (2015), ML trees based on partial cpDNAs and nucleoplasmic matrices of 29 *Chrysosplenium* species all indicated that *C. microspermum* was located at the base of the *Chrysosplenium*, but BI trees of the nucleoplasmic matrix did not show consistent results, with *C. microspermum* clustering with other alternate leaf species [18]. In Folk et al. (2019), although the Saxifragales phylogeny was analyzed using 301 phylogenetic loci, but the molecular data of *Chrysosplenium* was primarily based on partial chloroplast DNA data from previous studies [19]. This phylogeny showed that *C. microspermum* was clustered with other opposite leaf species and that *C. sedakowii* was located at the base of the *Chrysosplenium*. In this study, we provide more accurate support for the phylogeny of *C. microspermum* based on the cp genome and nrDNA data of 44 *Chrysosplenium* species. Our results support that *C. microspermum* was located at the base of the *Chrysosplenium* and was more closely related to *C. sedakowii*. This further showed that two sections of the *Chrysosplenium* were not monophyletic with each other.

Phylogenetic differences between nucleoplasm and between gene fragments may be due to various reasons, such as hybridization, incomplete lineage sorting, chloroplast capture, and plastid genetic differences. Hybridization occurs frequently in nature. Hybridization has the potential to result in gene trees that are inconsistent with species trees. Many naturally occurring hybrids, including intergeneric hybrids, have been reported in the family Saxifragaceae. Previous studies have suggested that hybridization occurs mainly in intergeneric hybrids between *Heuchera* and *Tiarella*, *Tellima* and *Tolmiea*, *Mitella* and *Conimitella*, and interspecific hybrids in *Heuchera* [45,46,47,48]. No hybridization events have been reported in *Chrysosplenium* species, but we cannot exclude the possibility of hybridization here. Incomplete lineage sorting is prevalent in most species phylogenies, where different fragments in the genome have different rates of evolution and conservation. The phylogenetic relationships constructed by different segments may differ somewhat from the true phylogenetic relationships and may also be related to chloroplast capture events. In the family Saxifragaceae, the *Tiarella* branch has been reported to have an apparent incongruity in the nucleoplasmic phylogeny, and the main reason for this incongruity was due to the fact that the *Tiarella* branch has captured at least two *Heuchera* cp genome events through an ancient ancestral hybridization [49]. In contrast, the phylogenetic trees for both nuclear and plastid phylogenies indicated that *Chrysosplenium* belonged to a monophyletic group, which was less likely for chloroplast capture of *Chrysosplenium* species with other genera, whereas it was possible to have chloroplast capture events within *Chrysosplenium*. Genetic differences in plastids may also have an effect on phylogeny, but this has not been reported in *Chrysosplenium* species. It is widely recognized that plastids are generally inherited matrilineally, but organelle genomes can also be mediated by biparental inheritance in the process of plant evolution. For example, *Medicago truncatula* and *Pelargonium zonale* exhibit frequent biparental inheritance [50,51]. Even in plants that are predominantly maternally inherited, such as *Nicotiana tabacum* and *Arabidopsis thaliana*, plastid genomes are occasionally inherited through pollen dispersal (paternal leakage) [52,53]. However, the causes and determinants of uniparental inheritance of organelles, as well as the underlying mechanisms of maternal inheritance, remain largely unknown. Previous cytological mechanisms of paternal inheritance of plastids have shown that mild low-temperature stress promotes the entry of paternal plastids into spermatocytes during male gametogenesis and significantly increases the inheritance of paternal plastids [54]. *Chrysosplenium* species prefer low temperature and low light environments, and it is possible that paternal plastid inheritance could be increased under low-temperature conditions, but further studies are needed. Therefore, in the future, there is a need not only to collect more *Chrysosplenium* species, but also to study a number of aspects such as nuclear and mitochondrial genomes, population genetics, plastid inheritance patterns, and chloroplast capture, in order to explore more accurate phylogenetic relationships within the genus and to construct a more believable phylogenetic network of the *Chrysosplenium*.

## 4. Materials and Methods

### 4.1. Sampling, DNA Extraction, and Sequencing

A total of 44 *Chrysosplenium* species covering the major continents of the world were used, of which 34 species of them were newly sequenced in this study. Species specimen number and collection location are listed in Appendix A. Leaf tissue was dried in silica gel and genomic DNA was extracted using the modified CTAB method [55]. Whole genome resequencing was performed at Biomarker Technologies Company in Beijing, China. The short insertion library was constructed, and then 2 × 150 bp paired-end reads were obtained from the Illumina NovaSeq platform. The adaptors and low-quality reads were removed using Trimmomatic v. 0.39 [56], and then the filtered reads were quality-controlled using Fastqc v. 0.11.9 [57].

### 4.2. Chloroplast Genome Assembly and Annotation

The cp genomes of *Chrysosplenium* were assembled from clean short reads using GetOrganelle v.1.7.5 [58]. The assembly parameters used were “-R 20 -k 21, 45, 65, 85, 105, 127 -F embplant_pt”. Then, we used Bandage to check the integrity of the genomes. The cp genome annotation was performed using CPGAVAS2 [59], PGA [60] and Geneious Prime v. 2022.2.2 [61]. Protein-coding genes (PCGs) were extracted using PhyloSuite v. 1.2.3 [62]. The cp genome map was constructed using CPGview (http://www.1kmpg.cn/cpgview/ (accessed on 10 April 2023)) [63]. For the nrDNA sequences, we also used GetOrganelle v.1.7.5 [58] to assemble them, setting the parameter “-R 7 -k 35, 85, 115 -F embplant_nr” and then annotated with Geneious Prime v. 2022.2.2 [61].

### 4.3. Repeat Structure Identification

Simple sequence repeats (SSRs) were identified using the MicroSatellite (MISA) [64]. The minimum repeat number was set at 10, 5, 4, 3, 3, and 3 for mononucleotide, dinucleotide, trinucleotide, tetranucleotide, pentanucleotide, and hexanucleotide, respectively. REPuter (https://bibiserv.cebitec.uni-bielefeld.de/reputer (accessed on 10 April 2023)) was used to count the long repeats of the cp genomes, including palindrome sequences and interspersed repeats (complement repeats, forward repeats and reverse repeats) [65]. The minimum repeat and hamming were set to 30 and 3, respectively.

### 4.4. Codon Usage Analysis

To reduce sampling error, we excluded protein-coding genes (PCGs) shorter than 300 bp when analyzing codon usage patterns. A total of 53 CDSs were used for codon usage analysis. We utilized CodonW v.1.4.4 to determine the GC of the silent 3rd codons, effective number of codons, codon adaptation index, and number of synonymous codons. Additionally, we employed PhyloSuite v1.2.3 [62] to calculate the relative synonymous codon usage (RSCU) value. An RSCU value greater than 1 indicates higher frequency of codon usage than expected, while an RSCU value less than 1 indicates lower frequency of codon usage than expected.

### 4.5. Sequence Variation Analysis

The cp genomes of 44 *Chrysosplenium* species were compared using mVISTA in shuffle LAGAN mode, and *C. ramosum* were used as a reference. Multiple sequence alignment was performed using MAFFT. The DnaSP v. 6.12.03 [66] was used to calculate the nucleotide diversity (Pi) of the cp genome by using the sliding window. The step and window size were set to 200 bp and 600 bp, respectively. The LSC-IRa, IRa-SSC, SSC-IRb, and IRb-LSC boundaries of 44 cp genomes of *Chrysosplenium* were visualized using IRscope. In addition, mauve and AliTV [67] were used to detect genomic rearrangement events. We also compared the chloroplast genomes of two different taxa of *C. sinicum* using Geneious Prime v. 2022.2.2 [61].

### 4.6. Selective Pressure Analysis

The 74 cp PCGs of 44 *Chrysosplenium* species and one *Peltoboykinia* species were used to evaluate evolutionary rate variation. Positive selection analysis was performed in four parts, namely within *Alternifolia* branch, within *Oppositifolia* branch, between *Alternifolia* and *Oppositifolia*, between *Chrysosplenium* and *Peltoboykinia*. KaKs_Calculator v. 2.0 with YN model was used to determine the ratio of non-synonymous substitutions (*K*a) and synonymous substitutions (*K*s) [68]. *K*a/*K*s < 1 indicates that the gene may be under purifying selection. *K*a/*K*s > 1 indicates that the gene may be under positive selection. *K*a/*K*s = 1 indicates that the gene may be under neutral selection. When *K*s = 0, the value of *K*a/*K*s is represented by NA, indicating that the gene has few nonsynonymous sites/substitutions. 

We also used the branch-site model in EasyCodeML [69] to further detect the positive selection sites of genes. For the positive selection prediction between *Chrysosplenium* and *Peltoboykinia*, we set the *Chrysosplenium* branch as the foreground branch and the *Peltoboykinia* branch as the background branch. And within *Chrysosplenium*, we used the *Oppositifolia* branch as the foreground branch and the *Alternifolia* branch as the background branch.

### 4.7. Phylogenetic Analysis

We selected 19 species as outgroups for phylogenetic analysis (Appendix A). Then, 74 common cpPCGs were extracted from the cp genome using PhyloSuite v.1.2.3 [62]. The 74 cpPCGs and nrDNA sequences were aligned separately using MAFFT v.7.4 [70], and then concatenated using PhyloSuite v.1.2.3 [62] to form a cpPCGs matrix, a cpPCGs + nrDNA matrix, and an nrDNA matrix. The phylogenetic tree was conducted using Maximum likelihood (ML) and Bayesian inference (BI) methods, respectively. ModelFinder [71] was used to find the best-fitting model for ML analysis, and the ML tree was further conducted using IQ-TREE v. 2.1.2 [72] with 1000 bootstrap replicates. For the BI tree, we used MrBayes v. 3.2.6 [73] to generate a maximum clade credibility (MCC) tree. The parameters were set as follows: nst = 6, rates = invgamma. The BI tree was performed with the concatenated sequence, using one million generations, two runs, four chains, a temperature of 0.001, and 25% of trees were discarded as burn-in, and trees were sampled every 1000 generations. The resulting tree was visualized using Figtree v. 1.4.4 (https://github.com/rambaut/figtree/Releases (accessed on 4 May 2023)).

## 5. Conclusions

In this study, we comprehensively performed assembly, comparative genomic, and phylogenetic analyses of multiple *Chrysosplenium* cp genomes. The analyses revealed that *Chrysosplenium* species were more conserved in terms of genome structure, gene content and arrangement, SSRs, and codon preference, but differ in genome size and SC/IR boundaries. Phylogenomic analysis showed that plastid data could effectively improve the phylogenetic support and resolution of *Chrysosplenium* species, strongly supporting *Chrysosplenium* as a monophyletic taxon and its internal division into three clades. The C. microspermum was not clustered with other *Chrysosplenium* species with alternate leaves but was clustered with *C. sedakowii* as the basal branch of *Chrysosplenium*. In addition, ten mutation hotspot regions were identified, which can be used as potential DNA barcodes for *Chrysosplenium* species identification. The *clp*P and *ycf*2 genes were significantly positively selected in the cp genome of *Chrysosplenium* compared to *Peltoboykinia* and had multiple positive selection sites of significance. One significant positive selection site was also detected in the *pet*G gene between the two sections. These positive selection sites may have played an important role in the evolutionary history of the *Chrysosplenium* species for their low-light adaptation. In conclusion, this study enriches the cp genomes of the *Chrysosplenium* species and provides a reference for subsequent studies on its evolution and origin.

## Figures and Tables

**Figure 1 ijms-24-14735-f001:**
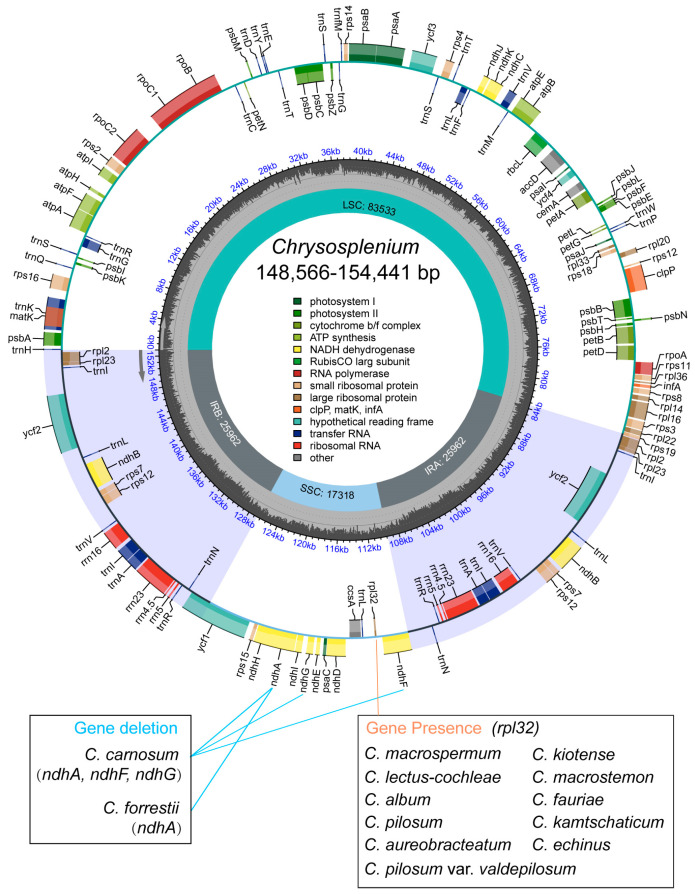
Representative chloroplast genome map of *Chrysosplenium*. The colored boxes in the figure represent genes. Genes located inside the circle are transcribed in a clockwise direction, while genes outside the circle are transcribed in a counter-clockwise direction. The small grey bar graphs in the inner circle indicate the GC contents. Black boxes indicate the absence or presence of individual genes in some *Chrysosplenium* species.

**Figure 2 ijms-24-14735-f002:**
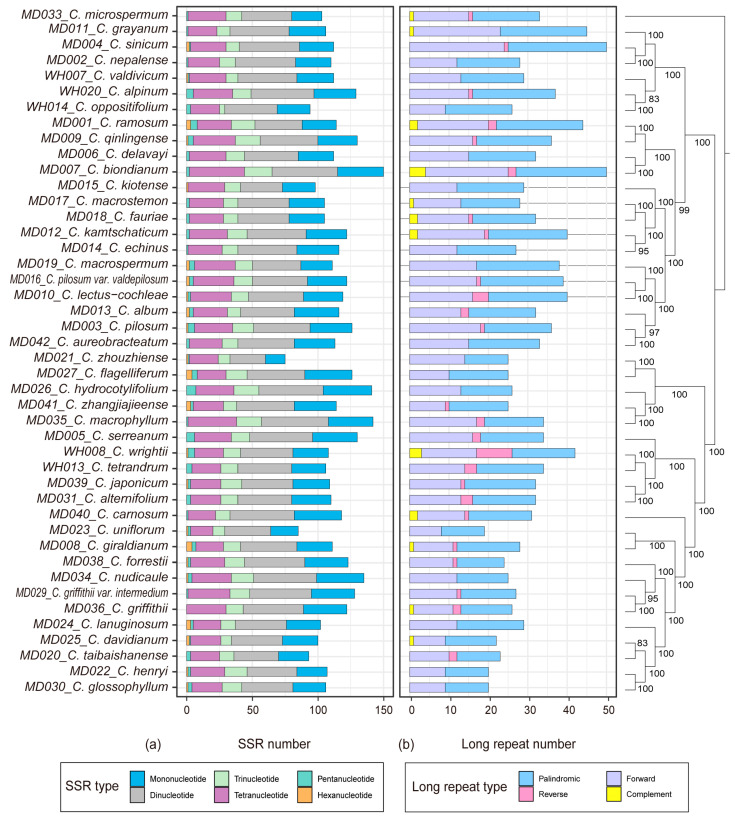
Repeat analysis of chloroplast genomes of *Chrysosplenium* species. (**a**) SSR statistics of *Chrysosplenium* species. Different types of SSRs are indicated by different colors. (**b**) Long repeat statistics of *Chrysosplenium* species. Different types of repeats are indicated by different colors. The values on the nodes indicate the ML bootstrap support values.

**Figure 3 ijms-24-14735-f003:**
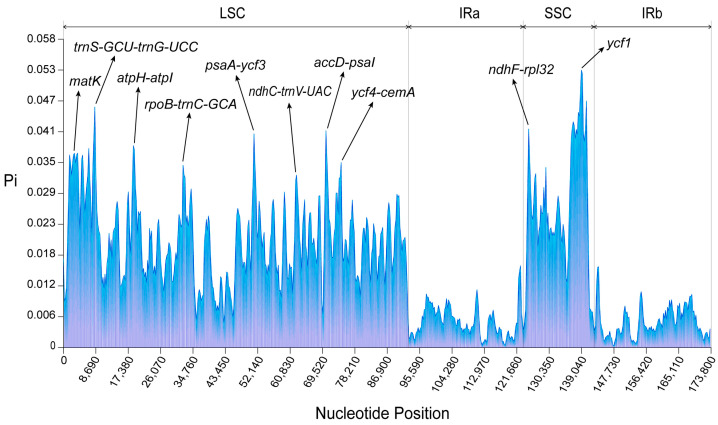
Nucleotide diversity (Pi) analysis of cp genomes of 44 *Chrysosplenium* species. The sliding window and step size used for this analysis were set to 600 bp and 200 bp, respectively.

**Figure 4 ijms-24-14735-f004:**
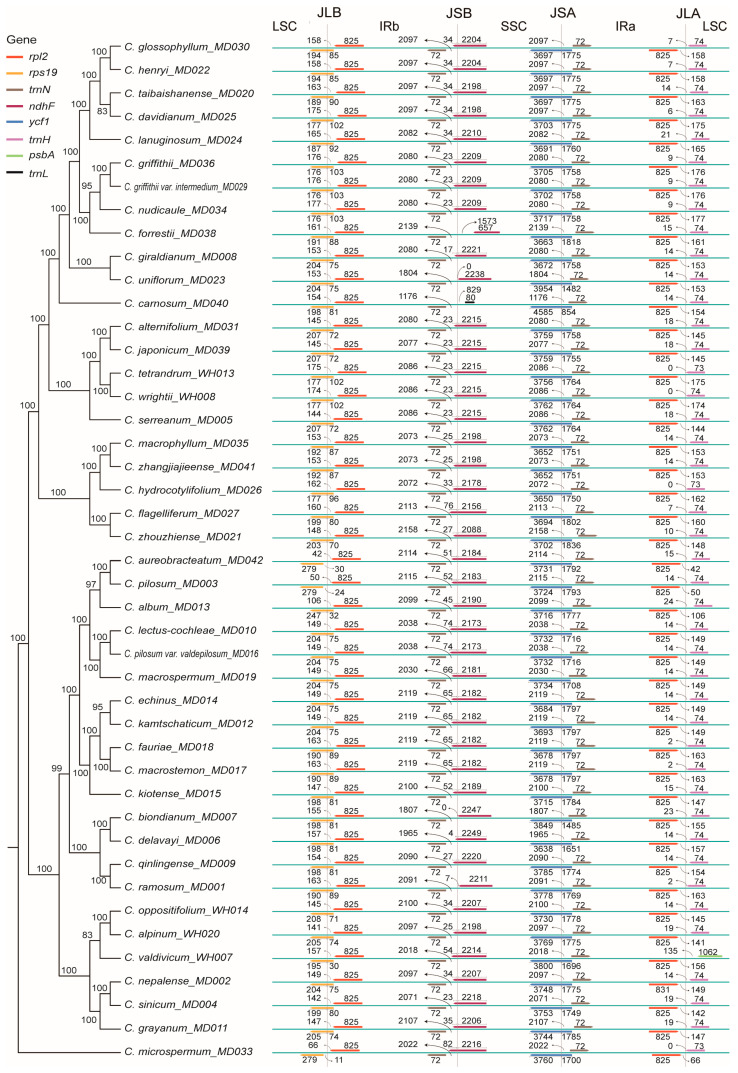
Dynamic analysis of the IR boundary of cp genomes of the 44 *Chrysosplenium* species. The values on the nodes indicate the ML bootstrap support values. Arrows indicate the distance of these genes from the IR boundary.

**Figure 5 ijms-24-14735-f005:**
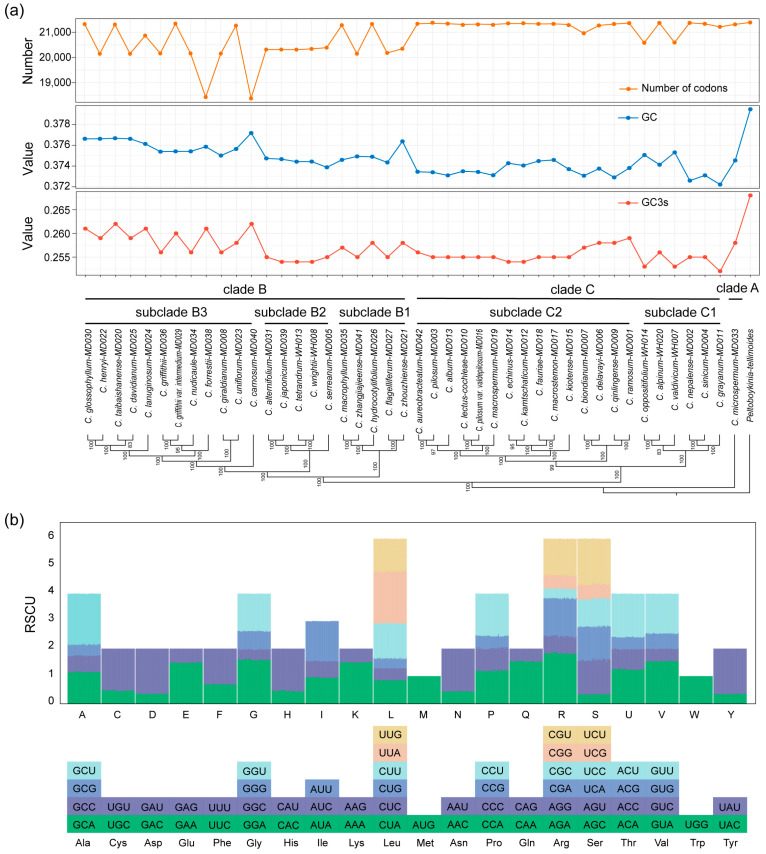
Codon characterization of PCGs in the cp genomes of 44 *Chrysosplenium* species. (**a**) Number of codons used, GC3s and GC content analysis. The values on the nodes indicate the ML bootstrap support values. (**b**) Codon preference (RSCU) analysis.

**Figure 6 ijms-24-14735-f006:**
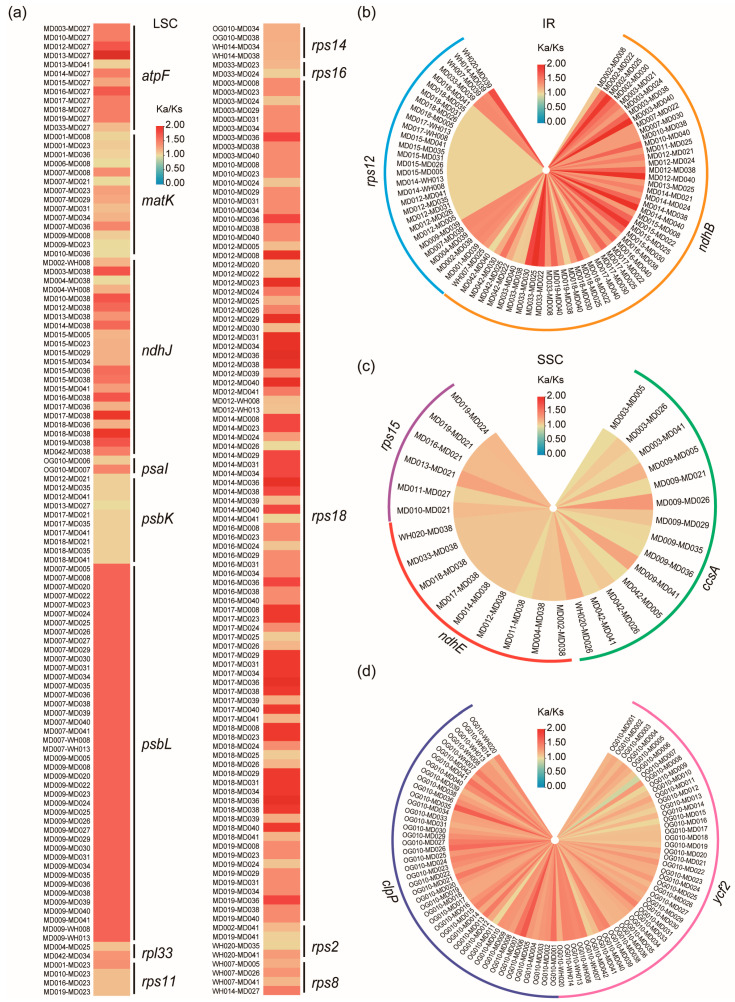
Selection pressure analysis of the genus *Chrysosplenium*. (**a**) Positive selection genes between *Alternifolia* and *Oppositifolia* and their species combinations in the LSC region. (**b**) Positive selection genes between *Alternifolia* and *Oppositifolia* and their species combinations in the IR region. (**c**) Positive selection genes between *Alternifolia* and *Oppositifolia* and their species combinations in the SSC region. (**d**) Positive selection genes *ycf2* and *clp*P between *Chrysosplenium* and *Peltoboykinia* and their species combinations.

**Figure 7 ijms-24-14735-f007:**
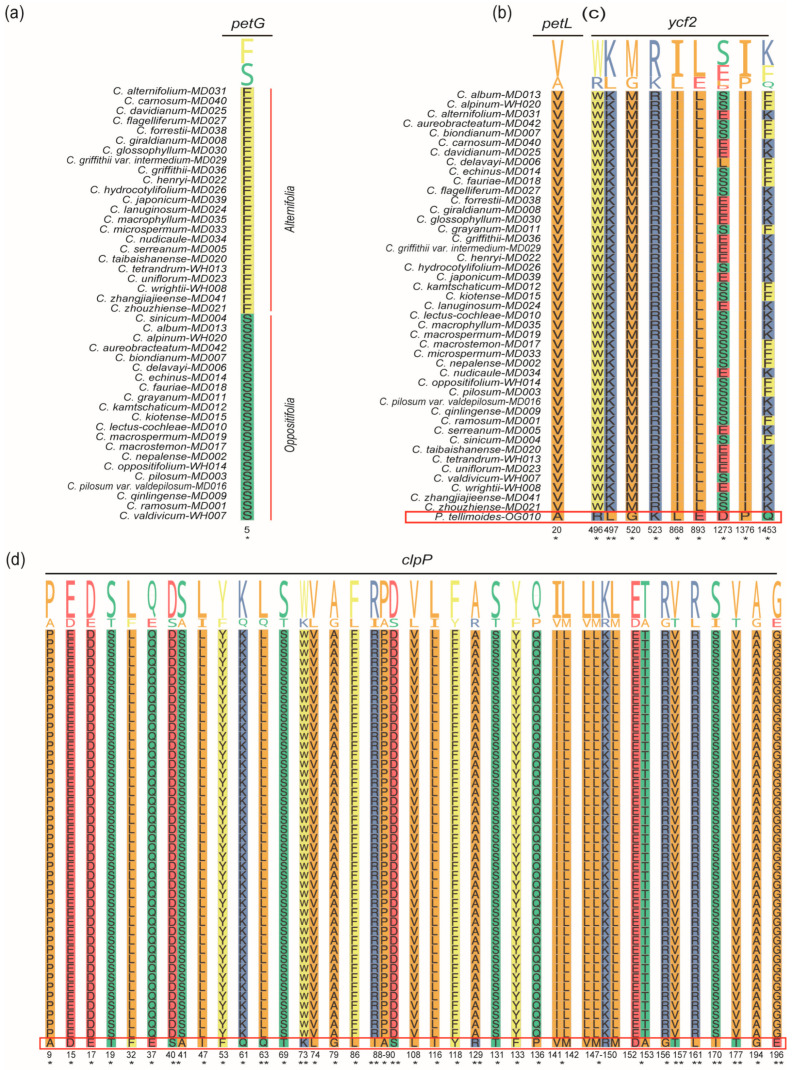
Analysis of positive selection sites in PCGs in the cp genome. (**a**) Positive selection sites between *Alternifolia* and *Oppositifolia* in the genus *Chrysosplenium*. (**b**–**d**) Positive selection sites between *Chrysosplenium* and *Peltoboykinia*. One asterisk indicates significance level less than 0.05; two asterisks indicate significance level less than 0.01.

**Figure 8 ijms-24-14735-f008:**
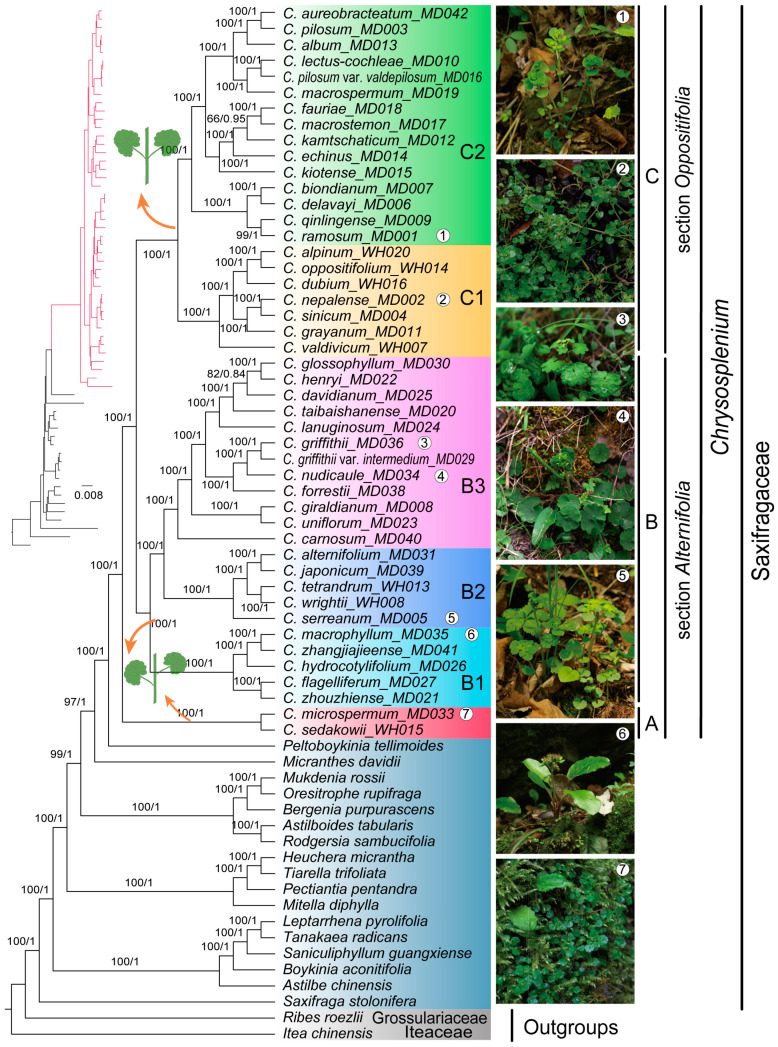
Phylogenetic tree of *Chrysosplenium* species using Maximum likelihood (ML) and Bayesian inference (BI) based on cpPCGs + nrDNA matrix. The values on the nodes indicate the ML bootstrap support values (**left**) and BI posterior probabilities (**right**). The circle numbers in the species picture correspond to the circle numbers behind the species name, respectively.

## Data Availability

The data provided in the study are deposited in GenBank database (https://www.ncbi.nlm.nih.gov/ (accessed on 20 June 2023)) and youdao cloud note (https://note.youdao.com/s/Uf9wZj68 (accessed on 4 August 2023)). Vouchers and GenBank accession numbers are listed in Appendix A.

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
