# Peer review of "A Comprehensive Analysis of Chloroplast Genome Provides New Insights into the Evolution of the Genus Chrysosplenium"

_ijms, 2023, doi:10.3390/ijms241914735_

Round 1

Reviewer 1 Report

The authors report chloroplast genomic sequences for 34 species of Chrysoplenium.  These sequences, in addition to 10 others previously available in the literature, are extensively characterized and then used to develop a phylogenetic hypothesis for relationships among the 44 species.  The characterization of the cp DNA sequences is excellent and provides a very good understanding of the variation and, to a certain extent, evolution of this molecule in the genus.  Shortcomings of the research include a lack of intraspecific variation in cp DNA and a serious analysis of potential effects of interspecific hybridization or reticulate evolution on their conclusions (the authors only mention these complicating factors in the last paragraph of the Discussion).  In addition, the possibility of paternal contribution of cp DNA is not discussed.  Although the Saxifragaceae displays primarily maternal inheritance of cp DNA, is there any study that has documented the percentage of paternal/biparental inheritance of cp DNA in Chrysoplenium?

The primary issue I found with the current manuscript is that many topics presented in the Introduction are not germane to the data presented in the Results or topics of focus in the discussion.  The first sentence of the Introduction is too general, and the authors should begin with the second sentence.  The description made on lines 39-41 is not true for all angiosperms and should be restricted to the Saxifragaceae.  The sentence on line 43 is overstated; cp DNA sequence analysis had made important contributions to plant systematics but has not ‘transformed this field’ (note that the authors comment later that morphological and DNA sequence data are equally important).

The second paragraph (lines 47-70) appears to assume that species are static, well-defined entities rather than evolving groups of populations.  I believe the Darwinian revolution rejected this concept.  Thus, any specific ‘bar code’ or sequence cannot be expected to unequivocally identify a species except in somewhat unusual (and temporary) circumstances.  As the manuscript does not make a major effort to use cp DNA bar codes to define Chrysosplenium species, I recommend deletion of this paragraph.

The following three paragraphs can be considerably condensed.  Comments regarding the ‘medicinal’ aspects of the genus can be omitted as this topic is not addressed in the rest of the manuscript.  I suggest a focus on what might be the most generally accepted phylogeny for the genus and where phylogenetic relationships still remain unresolved.  The three questions posed at the end of the Introduction are helpful in framing the discussion and conclusions, but fail to encompass the most significant findings of the research.

Other minor points:

Line 158: define ‘PCGs’ (authors do so in the M&M section, but this occurs later in the manuscript)

Lines 176-177:  Why mention species with lowest and greatest number of SSRs?  Such information can be obtained from figures and is not crucial to the manuscript.

Lines 183-184:  Same question as previous comment.

Figure 4 seems repetitive and could be omitted

Line 248:  start sentence with text rather than Arabic number (e.g. ‘Twenty-nine’)

Line 324: modify to ‘many angiosperm genera.’

Lines 332-334:  need references for these statements.

Lines 359-360:  Is ‘relatively fewer’ based on absolute number or a per nucleotide percentage?

Lines 362-382:  These speculations are not based on direct data.  The authors may want to further emphasize the unsupported nature of these possibilities

Lines 402-406:  the authors emphasize that there are inconsistencies between the morphological and DNA sequence data sets, but appear to minimize the inconsistencies among the molecular datasets.  Could the authors address the question why their findings differ from those of Soltis et al. and Folk et al?

Lines 416-417:  Why do the authors feel that more sequence data will resolve the current inconsistencies in Chrysosplenium phylogeny?  If hybridization and incomplete lineage sorting are interfering, a different approach will be necessary, not more of the same.

-

Author Response

Dear Reviewers:

Thank you for your letter and for the reviewers’ comments concerning our manuscript entitled “A comprehensive analysis of chloroplast genome provides new insights into the evolution of the genus Chrysosplenium” (ijms-2598599). Those comments are all valuable and very helpful for revising and improving our paper, as well as the important guiding significance to our researches. We have studied comments carefully and have made correction which we hope meet with approval. Questions are indicated by Q and responses are indicated by A. Modified sections in the article are shown in yellow.

Reviewer #1:

Q1. The authors report chloroplast genomic sequences for 34 species of Chrysoplenium. These sequences, in addition to 10 others previously available in the literature, are extensively characterized and then used to develop a phylogenetic hypothesis for relationships among the 44 species. The characterization of the cp DNA sequences is excellent and provides a very good understanding of the variation and, to a certain extent, evolution of this molecule in the genus. Shortcomings of the research include a lack of intraspecific variation in cp DNA and a serious analysis of potential effects of interspecific hybridization or reticulate evolution on their conclusions (the authors only mention these complicating factors in the last paragraph of the Discussion). In addition, the possibility of paternal contribution of cp DNA is not discussed. Although the Saxifragaceae displays primarily maternal inheritance of cp DNA, is there any study that has documented the percentage of paternal/biparental inheritance of cp DNA in Chrysoplenium?

A1. We thank the reviewers for their valuable suggestions and we have revised them accordingly. As for intraspecific variation in cp DNA of Chrysoplenium, compared analysis has been performed for the species and the corresponding variation. Differences also exist between genomes within species, a common phenomenon that may be due to genetic variation and geographic distribution during the evolution of the species. Similar differences exist in the genus Chrysosplenium, mainly in the gene spacer region, such as C. sinicum (Figure S6). However, there is still a lack of a more comprehensive resolution of genomic differences among intraspecific species in the genus Chrysosplenium. In future study, we will perform identify the intraspecific variation at the population level according to your suggestion.

As for the effect of interspecific hybridization on phylogeny, we have added the related content in the discussion section. The detailed descriptions were as follows. Hybridization has the potential to result in gene trees that are inconsistent with species trees. Many naturally occurring hybrids, including intergeneric hybrids, have been reported in the family Saxifragaceae. Previous studies have suggested that hybridization occurs mainly in intergeneric hybrids between Heuchera and Tiarella, Tellima and Tolmiea, Mitella and Conimitella, and interspecific hybrids in Heuchera. No hybridization events have been reported in Chrysosplenium species, but we cannot exclude the possibility of hybridization here. Incomplete lineage sorting is prevalent in most species phylogenies, where different fragments in the genome have different rates of evolution and conservation. The phylogenetic relationships constructed by different segments may differ somewhat from the true phylogenetic relationships and may also be related to chloroplast capture events. In the family Saxifragaceae, the Tiarella branch has been reported to have an apparent incongruity in the nucleoplasmic phylogeny, and the main reason for this incongruity was due to the fact that the Tiarella branch has captured at least two Heuchera cp genome events through an ancient ancestral hybridization. In contrast, the phylogenetic trees for both nuclear and plastid phylogenies indicated that Chrysosplenium belonged to a monophyletic group, which was less likely for chloroplast capture of Chrysosplenium species with other genera, whereas it was possible to have chloroplast capture events within Chrysosplenium.

As for the possibility of paternal contribution of cp DNA, we also have added the related content in the discussion section according to your suggestion. Genetic differences in plastids may also have an effect on phylogeny, but this has not been reported in Chrysosplenium species. It is widely recognized that plastids are generally inherited matrilineally, but organelle genomes can also be mediated by biparental inheritance in the process of plant evolution. For example, Medicago truncatula and Pelargonium zonale exhibit frequent biparental inheritance. Even in plants that are predominantly maternally inherited, such as Nicotiana tabacum and Arabidopsis thaliana, plastid genomes are occasionally inherited through pollen dispersal (paternal leakage). However, the causes and determinants of uniparental inheritance of organelles, as well as the underlying mechanisms of maternal inheritance, remain largely unknown. Previous cytological mechanisms of paternal inheritance of plastids have shown that mild low-temperature stress promotes the entry of paternal plastids into spermatocytes during male gametogenesis and significantly increases the inheritance of paternal plastids. Chrysosplenium species prefer low-temperature, low-light environments, and it is possible that paternal plastid inheritance could be increased under low-temperature conditions, but further studies are needed.

Q2. The primary issue I found with the current manuscript is that many topics presented in the Introduction are not germane to the data presented in the Results or topics of focus in the discussion. The first sentence of the Introduction is too general, and the authors should begin with the second sentence. The description made on lines 39-41 is not true for all angiosperms and should be restricted to the Saxifragaceae. The sentence on line 43 is overstated; cp DNA sequence analysis had made important contributions to plant systematics but has not ‘transformed this field’ (note that the authors comment later that morphological and DNA sequence data are equally important).

A2. Thanks for your critical suggestion. To make our topic concise, we have made the point-point revisions based on your suggestions.

Q3. The second paragraph (lines 47-70) appears to assume that species are static, well-defined entities rather than evolving groups of populations. I believe the Darwinian revolution rejected this concept. Thus, any specific ‘bar code’ or sequence cannot be expected to unequivocally identify a species except in somewhat unusual (and temporary) circumstances. As the manuscript does not make a major effort to use cp DNA bar codes to define Chrysosplenium species, I recommend deletion of this paragraph.

A3. OK. We have removed this paragraph.

Q4. The following three paragraphs can be considerably condensed. Comments regarding the ‘medicinal’ aspects of the genus can be omitted as this topic is not addressed in the rest of the manuscript. I suggest a focus on what might be the most generally accepted phylogeny for the genus and where phylogenetic relationships still remain unresolved. The three questions posed at the end of the Introduction are helpful in framing the discussion and conclusions, but fail to encompass the most significant findings of the research.

A4. OK. We have removed the medicinal introduction section and subsequently adjusted these three paragraphs.

Q5. Line 158: define ‘PCGs’ (authors do so in the M&M section, but this occurs later in the manuscript)

A5. OK. We've made changes.

Q6. Lines 176-177: Why mention species with lowest and greatest number of SSRs?  Such information can be obtained from figures and is not crucial to the manuscript.

A6. OK. We have removed this sentence.

Q7. Lines 183-184: Same question as previous comment.

A7. OK. We have removed this sentence.

Q8. Figure 4: seems repetitive and could be omitted

A8. OK. We have removed it.

Q9. Line 248: start sentence with text rather than Arabic number (e.g., ‘Twenty-nine’)

A9. OK. We've made changes.

Q10. Line 324: modify to ‘many angiosperm genera.’

A10. OK. We've made changes.

Q11. Lines 332-334: need references for these statements.

A11. OK. We have added the appropriate references.

Q12. Lines 359-360: Is ‘relatively fewer’ based on absolute number or a per nucleotide percentage?

A12. OK. This sentence emphasizes absolute number, and we have modified it.

Q13. Lines 362-382: These speculations are not based on direct data. The authors may want to further emphasize the unsupported nature of these possibilities.

A13. OK. We decide to remove this part as it is not a major finding in this paper

Q14. Lines 402-406: the authors emphasize that there are inconsistencies between the morphological and DNA sequence data sets, but appear to minimize the inconsistencies among the molecular datasets. Could the authors address the question why their findings differ from those of Soltis et al. and Folk et al?

A14. We thank the reviewers for this valuable advice and we have revised it accordingly. Morphological and molecular data complement each other in the phylogeny, which is also applied in the identification of new Chrysoplenium species (Fu et al., 2021, Chrysosplenium sangzhiense (Saxifragaceae), a new species from Hunan, China. PhytoKeys). This paper focuses more on phylogenetic differences between molecular data, so we have added a related exploration. Molecular systematics also often has problems that cannot be solved. For example, the evolutionary relationships of taxa with short-term radiation evolution are currently not clearly related by molecular systematics. Due to incomplete lineage sorting, hybridization, chloroplast capture, horizontal gene transfer, plastid genetic differences, etc., the evolutionary relationships of many genes are inconsistent with the evolutionary relationships of species, which greatly affects the inference of molecular data. In fact, most of the extant taxa have experienced species explosion, and to a certain extent, completely clear phylogenetic relationships cannot be obtained, and often only some of the relationships are clear. The above is still based on the situation that the amount of data is sufficient, and the few molecular markers commonly used in molecular systematics at present are prone to lack of representativeness. In turn, there are many different methods of constructing phylogenetic trees, such as the selection of gene sets, the selection of representative species, the use of algorithms and software, nucleic acids or proteins or parsimonious nucleic acid sequences, and even different ways of combining multiple genes such as tandem and parallel. After all, the results of clustering methods are not really stable and it is easy to get different results, leaving one confused in front of an unknown evolutionary history. What is more reliable at the moment is the concordance part of the different tree-building methods, but most of this concordance part relies on morphology as well. These may also account for the differences in the phylogeny of the Chrysoplenium by different researchers.

Q15. Lines 416-417: Why do the authors feel that more sequence data will resolve the current inconsistencies in Chrysosplenium phylogeny? If hybridization and incomplete lineage sorting are interfering, a different approach will be necessary, not more of the same.

A15. Thanks a lot for this valuable advice. We are adding and revising accordingly in the discussion section. We realize that more molecular data may still face similar problems, and that subsequent studies on population genetics, uniparental inheritance of organelles, hybridization, and chloroplast trapping in Chrysosplenium species will be needed to shed more light on the phylogeny of Chrysosplenium.

Thanks for your reconsideration.

Yours sincerely,

Tiange Yang (E-mail: yangtge@163.com)

23 Sep., 2023

South-Central Minzu University

Reviewer 2 Report

Dear authors,

thank you for your work.

You propose a work in which you assemble the chloroplast genomes of 34 Chrysosplenium species and perform genomic and phylogenetic analyzes in comparison with data obtained by other working groups. Furthermore, phylogenetic analysis of plastids showed that plastid data effectively improved the phylogenetic support and resolution of the species under study.

Your work is well articulated, well written, with plenty of information in supplementary figures and tables. The latter should be described in greater detail so as to be able to define them as useful for the work itself (example in figure S2 there are stars at the nodes of the branches but the legend does not indicate what they represent).

Furthermore, in the main body of the version supplied, unfortunately the internal part of figure 5 and part of 7a and b are absolutely not legible. I would ask you to provide these images in better resolution or to make arrangements with the publishers for their greater readability.

Since the authors state that they want to answer three specific questions, I would recommend directing the discussion and/or conclusion section more towards the answers to those questions.

Thank you.

Best Regards.

Author Response

Dear Reviewers:

Thank you for your letter and for the reviewers’ comments concerning our manuscript entitled “A comprehensive analysis of chloroplast genome provides new insights into the evolution of the genus Chrysosplenium” (ijms-2598599). Those comments are all valuable and very helpful for revising and improving our paper, as well as the important guiding significance to our researches. We have studied comments carefully and have made correction which we hope meet with approval. Questions are indicated by Q and responses are indicated by A. Modified sections in the article are shown in yellow.

Reviewer #2:

Q1. Your work is well articulated, well written, with plenty of information in supplementary figures and tables. The latter should be described in greater detail so as to be able to define them as useful for the work itself (example in figure S2 there are stars at the nodes of the branches but the legend does not indicate what they represent).

A1. We have made more detailed changes to the figure notes.

Q2. Furthermore, in the main body of the version supplied, unfortunately the internal part of figure 5 and part of 7a and b are absolutely not legible. I would ask you to provide these images in better resolution or to make arrangements with the publishers for their greater readability.

A2. Due to the figure being too large, the font was too small overall. So we have re-optimized the figure.

Q3. Since the authors state that they want to answer three specific questions, I would recommend directing the discussion and/or conclusion section more towards the answers to those questions.

A3. Thank you very much for your suggestion, we have optimized the content in the discussion section.

Thanks for your reconsideration.

Yours sincerely,

Tiange Yang (E-mail: yangtge@163.com)

23 Sep., 2023

South-Central Minzu University

Reviewer 3 Report

The main goal of this work was to sequence, assemble, and characterize chloroplast genomes of 44 (10 previously published and 34 new) examples of Chrysosplenium species.  The authors found that the general genome structure was highly conserved, with most changes occurring in non-coding regions and often consisting of repeats of variable lengths.  The overall phylogeny based on this work indicated that the species are monophyletic, with 3 main internal clades.  The authors identified some variable hotspot regions that would be suitable for quick analysis and identification of future species/specimens.  Over, this work was an interesting study.  A few suggested changes are as follows:

Main concerns:

1.       There were some examples of cp gene loss for a few of the species chosen.  Are there nuclear genome data available for those species and if so, was there evidence of gene transfer to the nucleus?  If these data are not available then just a mention of the possibility is sufficient.  

2.       Were the proposed polymorphic sites for use as candidates for DNA barcodes validated/tested? 

Minor comments

3.       Some of the figures (as presented) are quite small.  In particular Figure 3, as presented, is rather small for panel a.  This reviewer suggests moving panel a to supplemental materials and keeping figure b as a main figure (enlarged).  Figure 5, as presented, is too small to read.  Please increase the size. 

4.       The alternate and opposite foliage types had a clear division for the petG gene (F vs S).  Is this gene known to have a role in leaf morphology? 

5.       Beginning in line 44, there is a statement that “for more than a century, genome based phylogeny . . . ,” this seems unfeasible given the genomes were not sequenced until 1977 and chloroplasts in the mid 1980s.  Please revise this minor (but important!) typo. 

6.       Line 75 mentions fruit petals.  This is an unusual term, please define. 

7.       A few statements would benefit from citations.  For example:

8.       Lines 51-58 on first generation plant barcoding.

9.       Lines 413-414 on conflict of phylogeny types

10.   Lines 332-33 on movement of genes to nucleus

11.   Lines 413-414 on conflict between nuclear and cp genome phylogenies

12.   Please avoid use of red/green as informative colors in figures, not all readers can see the difference. 

13.   Line 361 please change play to plays. 

14.   It would be nice to see some images of the species studied (not all 44!), perhaps one of each foliage arrangement. 

No major concerns.  A few small edits are needed (see comments for any suggested changes).  

Author Response

Dear Reviewers:

Thank you for your letter and for the reviewers’ comments concerning our manuscript entitled “A comprehensive analysis of chloroplast genome provides new insights into the evolution of the genus Chrysosplenium” (ijms-2598599). Those comments are all valuable and very helpful for revising and improving our paper, as well as the important guiding significance to our researches. We have studied comments carefully and have made correction which we hope meet with approval. Questions are indicated by Q and responses are indicated by A. Modified sections in the article are shown in yellow.

Reviewer #3:

Q1. There were some examples of cp gene loss for a few of the species chosen. Are there nuclear genome data available for those species and if so, was there evidence of gene transfer to the nucleus? If these data are not available then just a mention of the possibility is sufficient.

A1. Thank you very much for your advice. However, the corresponding chromosome-level nuclear genomes have not been published in the Chrysosplenium species, which largely hinders the study of gene transfer between the chloroplast and nucleus.

Q2. Were the proposed polymorphic sites for use as candidates for DNA barcodes validated/tested?

A2. The matK sequence has been recognized as a universal phylogenetic marker in Chrysosplenium (Such as Soltis et al. 2001), but other added partial markers have not been validated in Chrysosplenium despite being used in other species taxa, and we will need to follow up with further validation.

Q3. Some of the figures (as presented) are quite small. In particular Figure 3, as presented, is rather small for panel a. This reviewer suggests moving panel a to supplemental materials and keeping figure b as a main figure (enlarged). Figure 5, as presented, is too small to read. Please increase the size.

A3. We have decided to put Figure 3a in the attachment and use Figure 3b as the main figure. Figure 5 we have optimized.

Q4. The alternate and opposite foliage types had a clear division for the petG gene (F vs S).  Is this gene known to have a role in leaf morphology?

A4. In Arabidopsis thaliana, petG controls the components of the cytochrome bf6-f complex subunit 5, which mediates electron transfer between photosystem II (PSII) and PSI, cyclic electron flow around PSI, and state transitions. However, the role of petG genes in the regulation of leaf morphology remains unreported.

Q5. Beginning in line 44, there is a statement that “for more than a century, genome-based phylogeny . . .,” this seems unfeasible given the genomes were not sequenced until 1977 and chloroplasts in the mid-1980s. Please revise this minor (but important!) typo.

A5. Combining the comments of several previous reviewers, we have revised this sentence.

Q6. Line 75 mentions fruit petals. This is an unusual term, please define.

A6. In Pan (1986), the split petals of the capsule are also called the fruit petals. Based on the suggestions of several previous reviewers, we have removed the section on the taxonomic characterization of the genus Chrysosplenium.

Q7. A few statements would benefit from citations. For example:

A7. We have made changes as follows.

Q8. Lines 51-58 on first generation plant barcoding.

A8. Thank you very much for your suggestion. However, combining the comments of the previous reviewers, we have deleted this paragraph.

Q9. Lines 413-414 on conflict of phylogeny types.

A9. We have optimized the discussion of the phylogenetic part of the genus Chrysosplenium. This sentence cites the corresponding reference.

Q10. Lines 332-33 on movement of genes to nucleus

A10. We have cited references.

Q11. Lines 413-414 on conflict between nuclear and cp genome phylogenies

A11. We have optimized the discussion of the phylogenetic part of the genus Chrysosplenium. This sentence cites the corresponding reference.

Q12. Please avoid use of red/green as informative colors in figures, not all readers can see the difference.

A12. Thank you very much for your advice, we try to avoid red and green.

Q13. Line 361 please change play to plays.

A13. This section has been optimized.

Q14. It would be nice to see some images of the species studied (not all 44!), perhaps one of each foliage arrangement.

A14. OK, we've added pictures of the Chrysosplenium species as shown in Figure 8.

Thanks for your reconsideration.

Yours sincerely,

Tiange Yang (E-mail: yangtge@163.com)

23 Sep., 2023

South-Central Minzu University